# Cell-Type Specific Analysis of Selenium-Related Genes in Brain

**DOI:** 10.3390/antiox8050120

**Published:** 2019-05-05

**Authors:** Alexandru R. Sasuclark, Vedbar S. Khadka, Matthew W. Pitts

**Affiliations:** 1Department of Cell and Molecular Biology, John A. Burns School of Medicine, University of Hawaii, 651 Ilalo Street, Honolulu, HI 96813, USA; asasu@hawaii.edu; 2Bioinformatics Core in the Department of Complementary and Integrative Medicine, John A. Burns School of Medicine, University of Hawaii, 651 Ilalo Street, Honolulu, HI 96813, USA; vedbar@hawaii.edu

**Keywords:** selenium, selenoprotein, brain, RNAseq, ELENOP, SELENBP1

## Abstract

Selenoproteins are a unique class of proteins that play key roles in redox signaling in the brain. This unique organ is comprised of a wide variety of cell types that includes excitatory neurons, inhibitory neurons, astrocytes, microglia, and oligodendrocytes. Whereas selenoproteins are known to be required for neural development and function, the cell-type specific expression of selenoproteins and selenium-related machinery has yet to be systematically investigated. Due to advances in sequencing technology and investment from the National Institutes of Health (NIH)-sponsored BRAIN initiative, RNA sequencing (RNAseq) data from thousands of cortical neurons can now be freely accessed and searched using the online RNAseq data navigator at the Allen Brain Atlas. Hence, we utilized this newly developed tool to perform a comprehensive analysis of the cell-type specific expression of selenium-related genes in brain. Select proteins of interest were further verified by means of multi-label immunofluorescent labeling of mouse brain sections. Of potential significance to neural selenium homeostasis, we report co-expression of selenoprotein P (SELENOP) and selenium binding protein 1 (SELENBP1) within astrocytes. These findings raise the intriguing possibility that SELENBP1 may negatively regulate astrocytic SELENOP synthesis and thereby limit downstream Se supply to neurons.

## 1. Introduction

The brain is comprised of a variety of specialized cell types that are broadly grouped into three major categories: glutamatergic excitatory neurons, GABAergic inhibitory neurons, and glia. Historically, neural cells have been classified according to their location, morphology, electrophysiological properties and/or expression of molecular markers. Advances in single-cell RNA sequencing (RNAseq) provide a means to more definitively identify cell types based on a molecular census of the entire transcriptome. Recently, the Allen Brain Atlas (ABA) has added a freely accessible online Cell Types Database [1], allowing researchers to access RNAseq transcriptomic data compiled from single cells derived from the mouse and human cortex. This newly developed resource is invaluable for advancing understanding of the diverse mix of cell types that make up the brain. Herein, we utilized this tool to analyze the cell-type specific expression of selenium (Se)-related genes in the mouse and human brain.

Selenoproteins are an essential class of proteins involved in redox signaling with important roles in neural function. These proteins are characterized by the co-translational incorporation of selenium (Se) in the form of selenocysteine (Sec) and include the glutathione peroxidases (GPX), thioredoxin reductases (TXNRD), iodothyronine deiodinases (DIO), and a host of additional Sec-containing proteins with undetermined molecular functions [2,3]. Several of these proteins are indispensable for brain development, as revealed by mouse knockout studies. In particular, selenoproteins are required for the functionality of parvalbumin-expressing interneurons (PVIs), a specific class of neurons characterized by fast-spiking activity and high rates of metabolism [4,5,6,7]. Se delivery to neurons is primarily mediated by selenoprotein P (SELENOP) and its cognate receptor, low-density lipoprotein receptor-related protein 8 (LRP8, also known as ApoER2) [8,9]. SELENOP is distinctive among selenoproteins, due to the fact that its C-terminal domain contains multiple Sec residues that facilitate extracellular Se transport. Moreover, genetic ablation of SELENOP or LRP8 results in diminished brain Se levels [9,10,11] and severe neurological dysfunction upon administration of a Se-deficient diet [12,13]. We previously reported PVI deficits in *Selenop^−/−^* mice and showed that LRP8 expression is enriched on PVIs in certain brain regions, indicating that SELENOP may be preferentially targeted to this important and metabolically-active cell type [14]. However, whether individual selenoproteins are preferentially expressed by PVIs or other distinct cell types has yet to be investigated in a comprehensive manner.

In this study, we probed the cell-type specific expression of Se-related genes using single-cell RNAseq data obtained through the online ABA Cell Types Data Portal [1]. Analyses were conducted on 23,822 mouse and 15,928 human cortical cells that were originally grouped into three major classes (Non-neuronal, Glutamatergic, GABAergic) and further divided into distinct subclasses based upon their transcriptional profile.

## 2. Materials and Methods

### 2.1. Allen Brain Atlas RNAseq Analysis

The ABA Cell Types online data portal [1] allows researchers to freely access a host of electrophysiological, morphological, and transcriptomic data compiled from single cells. These cells are derived from cortical regions of the mouse and human brain. For this study, we analyzed RNAseq transcriptomic data for Se-related genes in 23,822 mouse cells and 15,928 human cells. The mouse cells were derived from the anterior lateral motor area (9573 cells) and the primary visual cortex (14,249 cells), whereas the human cells were all derived from the middle temporal gyrus. Human samples were obtained from 7 individuals, ranging from 24–66 years of age. Mouse samples were taken from 17 mice, ranging from 51–91 days of age. For both mouse and human, samples were obtained from both sexes. Additional details regarding tissue preparation, cell isolation, and RNA sequencing are described in the technical white paper on transcriptomics that is available at the ABA Cell Types data portal (http://help.brain-map.org/display/celltypes/Documentation).

We analyzed RNAseq data for 22 selenoproteins, which comprise all known selenoproteins common to mouse and man, with exception of SELENOH and SELENOV, for which data was not available. Selenoproteins examined were as follows: GPX1, GPX2, GPX3, GPX4, TXNRD1, TXNRD2, TXNRD3, DIO1, DIO2, DIO3, SELENOF (SEP15), SELENOI (EPT1), SELENOK (SELK), SELENOM (SELM), SELENON (SEPN1), SELENOO (SELO), SELENOP (SEPP1), MSRB1, SELENOS (VIMP), SELENOT (SELT), SELENOW (SEPW1), and SEPHS2. In addition to the aforementioned selenoproteins, we also probed for 12 additional Se-associated genes known to contribute to Sec biosynthesis/incorporation or Se metabolism/transport. Chosen genes involved in Sec biosynthesis/incorporation included: *SECISBP2*, *SEPSECS*, *TRNAU1AP*, *EEFSEC*, *EIF4A3*, *RPL30*, and *NCL*. Genes of interest related to Se metabolism/transport were: *LRP1*, *LRP2* (*Megalin*), *LRP8* (*ApoER2*), *SCLY*, and *SELENBP1*. For a more detailed overview of how these proteins influence selenium metabolism and selenoprotein synthesis, we refer our readers to several excellent review articles on this topic [2,15,16]. Heatmaps for selenoproteins and Se-associated genes were generated using log_10_ cpm values obtained from single cell RNAseq data downloaded from the ABA Cell Types online data portal [1]. Plots were created using heatmap.2 function in the gplots package in RStudio version 3.5.2. (RStudio, Boston, MA, USA)

Based upon the overall transcriptomic data (Appendix A), cells were initially grouped into 3 major classes: non-neuronal (Appendix A), glutamatergic (Appendix A), and GABAergic (Appendix A). Non-neuronal cells were further classified into 5 cell types, corresponding to: astrocytes, endothelial cells, microglia, oligodendrocytes, and oligodendrocyte precursor cells (OPCs). With respect to GABAergic neurons, ABA transcriptomic analysis of RNAseq data determined the presence of 61 distinct cell types in mouse and 45 in humans. To simplify matters for the purpose of this paper, GABAergic cells were subdivided into 4 cell types, as determined by expression of the largely non-overlapping marker genes: *LAMP5*, *PVALB*, *SST*, and *VIP*. With regard to glutamatergic neurons, ABA transcriptomic analyses of revealed 56 cell types in mice and 24 cell types in humans. For simplicity, we grouped glutamatergic neurons into 3 subclasses, as determined by expression of *FEZF2*, *PENK*, and *RORB*. Whereas the designated ABA subclasses among non-neuronal cells and gabaergic neurons were very similar between mouse and human, the glutamatergic subclasses were much more divergent between species. Hence, we set a cut-off value of 2.0 log_10_ cpm for the aforementioned marker genes (*FEZF2*, *PENK*, *RORB*) and filtered single-cell data for all glutamatergic cells. Select cells meeting the criteria mentioned above were allocated into the *FEZF2+, PENK+,* and *RORB+* glutamatergic subclasses.

### 2.2. Antibodies and Mice

The primary antibodies used for immunofluorescence were: rabbit anti-glial fibrillary acidic protein (GFAP) (1:2000; Z0334 DAKO, Carpinteria, CA, USA), rabbit anti-LRP8 (1:100; ab108208, Abcam, Cambridge, UK), rabbit anti-TXNRD1 (1:200; NBP1-96738, Novus, Littleton, CO, USA), mouse anti-SELENOM (1:200; sc-514952, Santa Cruz, Dallas, TEX, USA), rabbit anti-PV (1:2000; PV 27, Swant, Switzerland), mouse anti-PV (1:2000; PV 235, Swant), mouse anti-SELENBP1 (1:200; sc-373726, Santa Cruz), and rat anti-SELENOP (5 μg/mL; 9S4, Developmental Studies Hybridoma Bank, Iowa City, IA, USA). The SELENOP antibody targets the N-terminal domain and has been previously validated for immunohistochemistry in several published studies [17,18,19]. Tissue used for immunostaining was derived from adult male wild-type mice (*n* = 5) on a C57BL/6N background that were born and bred in the University of Hawaii Vivarium (Honolulu, HI, USA). All procedures and experimental protocols were approved by the University of Hawaii’s Institutional Animal Care and Use Committee (ethical code: A3423-01).

### 2.3. Multi-Label Immunofluorescent Microscopy

Mice were deeply anesthetized (1.2% avertin; 0.7 mL/mouse) and perfused intracardially with cold 0.1 M phosphate-buffer (PB) followed by 4% paraformaldehyde (PFA). Brains were removed, stored in 4% PFA for 24 h, immersed in graded solutions of sucrose (10%, 20%, 30%), and then cut into 40 µm coronal sections. Sections were treated with 0.3% H_2_O_2_ to inactivate endogenous peroxidases, blocked, and incubated with the proper primary antibody. For labeling with multiple primary antibodies, we employed a sequential approach where each individual primary antibody was incubated overnight at 4 ℃ and sections were probed with appropriate fluorescently-coupled secondary antibodies for visualization on the following day. Following labeling for antigens of interest, tissue was rinsed in phosphate buffered saline (PBS) and counterstained with NeuroTrace 640/660 (1:250; N21483, Thermo Fisher, Waltham, MA, USA) to visualize cell bodies. Sections were then mounted on slides with anti-fade fluorescent mounting media (Prolong Gold, Invitrogen, Carlsbad, CA, USA), and coverslipped. Slides were imaged using a Leica SP8 confocal microscope (Leica, Wetzlar, Germany) housed in the John A Burns School of Medicine Microscopy and Imaging core facility.

## 3. Results

### 3.1. General Profile of Se-Related Transcriptome in Cortical Cells

For this study, we analyzed the cell-type specific expression of 22 selenoproteins and 12 additional genes associated with Se metabolism, transport, and Sec biosynthesis in cortical cells derived from mice and humans. In order to first obtain a broad overview of the Se-related transcriptome, cells were initially grouped into three main classes: non-neuronal, glutamatergic, and GABAergic (Table 1; Appendix A). Separate heatmaps of gene expression were generated for mouse selenoproteins (Figure 1A), human selenoproteins (Figure 1B), mouse Se-associated genes (Figure 1C), and human Se-associated genes (Figure 1D). With the exception of *DIO2* and *SELENOP*, expression levels of selenoproteins were greater in neurons relative to non-neuronal cells. Neuronal selenoprotein expression was also generally higher in mouse, as 11 mouse selenoprotein genes (*GPX1, GPX4, TXNRD1, SELENOF, SELENOI, SELENOK, SELENOM, SELENOS, SELENOT, SELENOW, SEPHS2*) had mean values greater than 1.0 log_10_ cpm, whereas only 4 human genes (*GPX4, SELENOF, SELENOO, SELENOW*) were above this threshold. Interestingly, SELENOO was the only selenoprotein where neuronal expression was higher in humans relative to mouse. With respect to the Se-associated proteins, *SELENBP1* was predominantly expressed in non-neuronal cells, while levels of genes implicated in Se transport (*LRP8*), Se recycling (*SCLY*), and Sec biosynthesis/incorporation (*SECISBP2, TRNAU1AP, EEFSEC, EIF4A3, RPL30, NCL*) were markedly higher in neurons for both mice and humans.

### 3.2. Profile of Se-Related Transcriptome in Non-Neuronal Cells

We next examined the expression profile of Se-related genes within distinct populations of non-neuronal cells, dividing these cells into subclasses corresponding to astrocytes, endothelial cells, microglia, oligodendrocytes, and oligodendrocyte precursor cells (OPCs) (Appendix A). As noted above, of the 34 Se-related genes surveyed, only three (*DIO2, SELENOP, SELENBP1*) were found predominantly expressed in non-neuronal cells. Further analysis revealed that *DIO2* and *SELENBP1* were both largely produced by astrocytes (Figure 2). The expression profile of *SELENOP* was more nuanced, with levels being highest in astrocytes, endothelial cells, and microglia in mice, and within endothelial cells and oligodendrocytes in humans. We also observed enriched expression of *GPX1* in microglia relative to other cell types, in accord with prior findings in humans [20]. For mouse cells, several selenoproteins that were highly expressed in neurons (*GPX4, SELENOF, SELENOK, SELENOM, SELENOS, SELENOW*) were also broadly expressed at high levels (>1.0 log_10_ cpm) across multiple non-neuronal cell types, whereas in humans, only *SELENOW* was above this threshold. Finally, among non-neuronal cells, *LRP8* expression was greatest in endothelial cells, consistent with its known role in SELENOP uptake at the blood-brain barrier [21].

To further investigate SELENOP expression, we performed immunohistochemistry on mouse brain sections. SELENOP expression largely overlapped with that of GFAP+ astrocytes and was generally more prominent in white matter. For instance, we observed minimal SELENOP immunoreactivity in the mouse primary visual cortex, whereas levels were much more apparent in the adjacent dorsal hippocampal commissure (Figure 3A). We also performed immunohistochemistry in parallel for the selenium binding protein 1 (SELENBP1), a relatively uncharacterized protein that binds selenite [22,23] and is thought to modulate Se metabolism and redox status [24]. In accord with the RNAseq single-cell data, SELENBP1 expression was predominantly localized to astrocytes (Figure 3B) and was co-expressed with SELENOP (Figure 3C). We also found that SELENOP expression was most robust in the choroid plexus and regions lining the brain ventricles. Double-label immunofluorescence revealed extensive, but not complete, overlap of SELENOP with LRP8 in the choroid plexus (Figure 4A) and along the ventricular walls (Figure 4B,C). LRP8 was particularly abundant in ependymal cells, while SELENOP appeared to be largely expressed in adjacent astrocytes with endfeet projecting toward the ventricular lining.

### 3.3. Profile of Se-Related Transcriptome in Glutamatergic Neurons

In our initial profiling of the three primary cell classes, many Se-related proteins showed enriched expression in glutamatergic neurons. Those displaying elevated levels in both mouse and human cells corresponded to *TXNRD1, TXNRD2, EPT1, SELENOW, SEPHS2, SECISBP2*, and *EIF4A3* (Figure 1). For further analysis, glutamatergic neurons were divided into 3 subclasses, as determined by expression of the marker genes: *FEZF2, PENK, and RORB* (Appendix A). For the vast majority of genes surveyed, expression levels were relatively comparable across the three glutamatergic subclasses (Figure 5). Yet, of potential significance, heightened expression of *TXNRD2* and *SELENOI* was observed in *PENK^+^* cells relative to all other cell subclasses in both mice and humans.

### 3.4. Profile of Se-Related Transcriptome in GABAergic Neurons

We initially found that overall levels of *GPX4, SELENOK, SELENOM*, and *SELENOT* were highest in GABAergic cells among the main cell classes (Figure 1). To probe further, we divided GABAergic cells into 4 subclasses, as determined by expression of the marker genes, *LAMP5, PVALB, SST*, and *VIP* (Appendix A). Based upon prior evidence showing the necessity of selenoproteins for the viability of PVIs, we anticipated that selenoprotein levels may be higher in this class of interneurons. Consequentially, expression levels of selenoproteins and Se-associated genes were relatively comparable across GABAergic subclasses (Figure 6). However, it should be noted that levels of *TXNRD1, SELENOI*, and *SEPSECS* were marginally higher in PVIs relative to other GABAergic subclasses. These proteins appear to be critical for excitatory-inhibitory balance, as mutations in all of the aforementioned genes have been linked to epilepsy in humans [25,26,27]. To further investigate Se-related proteins in PVIs, we also conducted additional immunofluorescent staining for LRP8, SELENOM, and TXNRD1 (Figure 7). These proteins were broadly expressed across multiple neuronal subtypes, but levels appeared subtly enriched within PVIs.

## 4. Discussion

In summary, our results show that *DIO2, SELENOP*, and *SELENBP1* are predominantly expressed in non-neuronal cells, whereas the vast majority of Se-related proteins are most abundant in neurons. These findings are largely in line with that of an earlier study of selenoprotein expression in the mouse brain, which utilized the ABA to analyze expression in 159 discrete brain regions [28]. In addition to reporting that neurons represent the main sites of selenoprotein expression, this paper also found that the most abundant selenoproteins in mouse brain corresponded to *GPX4, SELENOF* (previously *SEP15*), *SELENOK, SELENOM, SELENOP*, and *SELENOW*. Likewise, with the exception of *SELENOP*, we found that all these genes were robustly expressed in mouse neurons and that *GPX4, SELENOF*, and *SELENOW* were highly expressed in human neurons as well. Interestingly, expression levels of *SELENOK, SELENOM*, and *SELENOS* were substantially lower in humans relative to that observed in mice, while levels of the poorly characterized *SELENOO* were elevated. Very little is known regarding *SELENOO*, but this raises the possibility that increased expression of this gene provided some benefit during the course of human evolution. In agreement with findings of Zhang et al., we also found that levels of most Se-associated genes were roughly 2-fold higher in neurons relative to glia in both mice and humans. The aforementioned paper also reported that selenoprotein expression was most pronounced regionally in the olfactory bulb, cerebral cortex, hippocampus, and cerebellar cortex. For the purposes of this study, we chose to focus upon the cerebral cortex and further investigate the cell-type specific expression of Se-related proteins.

Among selenoproteins known to contribute significantly to neural function, we found that *SELENOP* displayed a vastly distinct expression profile. High levels of *SELENOP* were observed in non-neuronal cells, whereas levels were barely detectable in neurons. We further verified SELENOP expression at the protein level by means of immunohistochemistry, observing co-expression in GFAP+ astrocytes. Our findings are concordant with prior studies reporting SELENOP expression in astrocytes [29,30], choroid plexus [21], and in ependymal cells lining the brain ventricles [31]. LRP8 was also particularly abundant in ependymal cells, lending further support to the notion that LRP8 participates in SELENOP uptake at the brain-cerebrospinal fluid barrier. Moreover, it also appeared that SELENOP expression was most pronounced in adjacent astrocytes, rather than the ependymal cells directly lining the ventricles. These findings are supportive of a model where SELENOP present in blood and CSF is taken up by LRP8-positive cells in the epithelial (blood-brain barrier) and ependymal (brain-CSF barrier) layers, resynthesized in neighboring astrocytes, and subsequently released to supply LRP8-positive neurons within the brain with Se (Figure 8). Furthermore, there is also a means by which Se can enter the brain independent of the LRP8-SELENOP pathway, as Se supplementation mitigates neurological dysfunction in *Selenop^−/−^* mice [13,32].

In addition to LRP8, another potential key player in brain Se metabolism is SELENBP1. This protein was initially identified in mouse liver by means of ^75^Se labeling [23] and later sequenced [33]. It has been shown to form a physical interaction with GPX1 and to attenuate GPX activity when overexpressed in cell culture studies [34]. More recently, SELENBP1 was demonstrated to catalyze the oxidation of methanethiol [35], generating H_2_O_2_ as a byproduct in the process. Although SELENBP1 upregulation has been reported in multiple postmortem schizophrenia studies [36,37,38], its functional role in brain remains unclear. Our observation of co-expression with SELENOP suggests that SELENBP1 may influence Se homeostasis within astrocytes. We hypothesize that cytosolic SELENBP1 may prevent Se from being incorporated into selenoproteins, thereby limiting SELENOP synthesis and downstream Se supply to neurons (Figure 8).

Given the broad scale of this study, several limitations merit acknowledgment. First, the cells examined in this study were derived from young adult mice and adult humans. Whereas our results pertain to the baseline adult state, it is highly probable that the cell-type specific expression profile of Se-related genes varies significantly during development, aging, and in certain disease states. For example, several selenoproteins have been found upregulated in astrocytes in response to brain injury and/or neurotoxicity, including TXNRD1 [39], GPX4 [40], and SELENOS [41,42]. Second, the dissected cortical regions used to generate single cells for RNAseq differed between mice and humans, as the human cells originated from the middle temporal gyrus and the mouse cells derived from both the anterior lateral motor area and primary visual cortex. Yet, regardless of this limitation, for the vast majority of Se-related proteins the cell-type specific expression was remarkably comparable between mice and humans. Finally, by definition, the RNAseq analysis does not measure protein levels. Whereas, levels of mRNA and protein are typically well correlated, post-transcriptional modifications add an additional layer of regulation that can greatly impact protein levels of any given gene. Given the critical role of selenoproteins for PVI viability, we originally speculated that levels of Se-related proteins might be substantially higher in this cell type. While this notion was not well supported by the RNAseq data, it remains possible that post-transcriptional mechanisms act to enhance protein synthesis and/or stability of selenoproteins within PVIs.

One other important variable that merits mention is sex, as sex-specific differences in Se homeostasis have been widely reported. We conducted additional analyses for the three main cell classes where individual cells were separated by sex and found that for the vast majority of genes, levels were comparable between sexes (Appendix A). However, it is worth noting that the selenoprotein with the most sexually dimorphic expression was SELENOP. Levels were higher in males, in line with prior studies showing that androgens stimulate SELENOP expression [43].

## 5. Conclusions

This study represents the first comparative analysis of the cell-type specific expression of Se-related proteins in the mouse and human brain. Our cumulative results are largely supportive of many prior studies that utilized immunohistochemistry to probe neural expression of individual selenoproteins. Of potential importance to neural Se homeostasis, we observed co-expression of SELENOP and SELENBP1 in astrocytes, raising the possibility that these two proteins work in tandem to regulate Se metabolism. Whether this proves to be significant is a matter for future investigation.

## Figures and Tables

**Figure 1 antioxidants-08-00120-f001:**
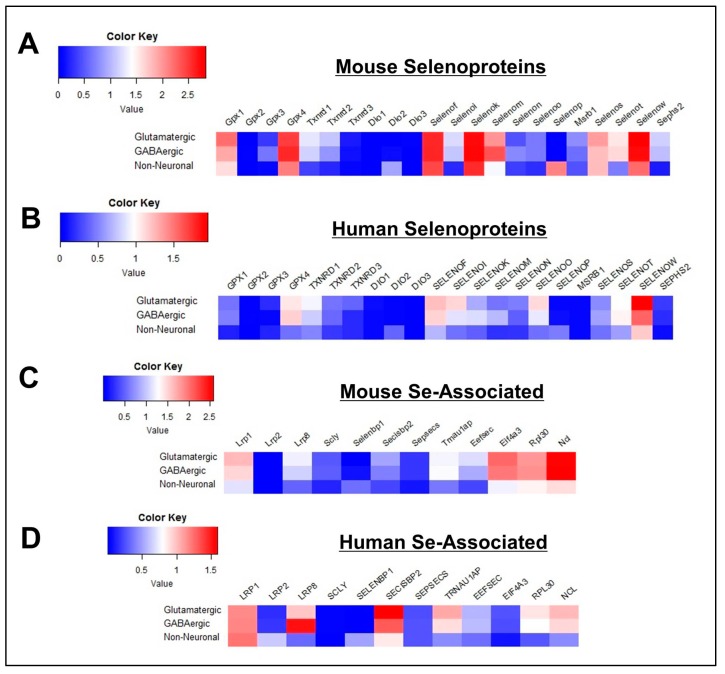
General profile of Se-Related Transcriptome in Cortical Cells. (**A**,**B**) Heatmaps detailing expression of mouse (**A**) and human (**B**) selenoprotein genes in glutamatergic, GABAergic, and non-neuronal cells. (**C**,**D**) Corresponding heatmaps of mouse (**C**) and human (**D**) Se-associated genes.

**Figure 2 antioxidants-08-00120-f002:**
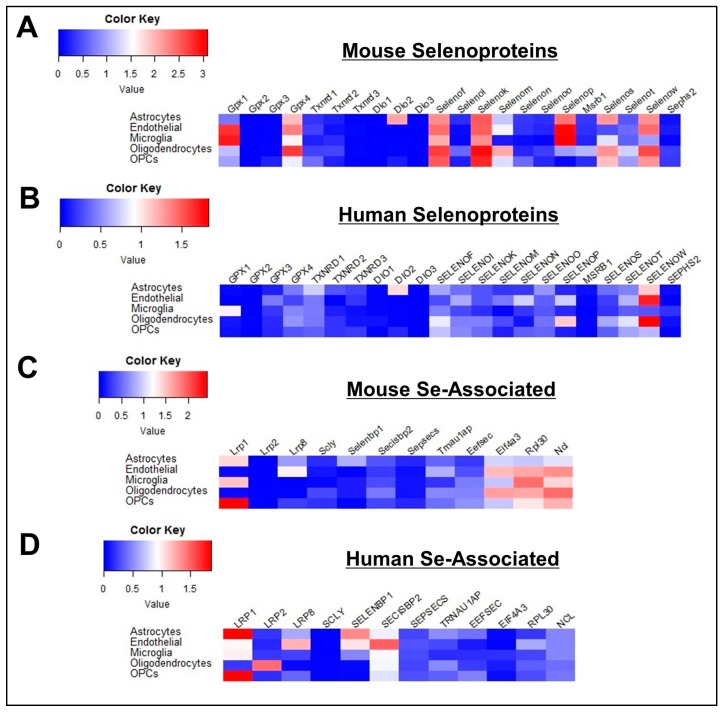
Profile of Se-Related Transcriptome in Non-Neuronal Cells. (**A**,**B**) Heatmaps detailing expression of mouse (**A**) and human (**B**) selenoprotein genes in astrocytes, endothelial cells, microglia, oligodendrocytes, oligodendrocyte precursor cells (OPCs). (**C**,**D**) Corresponding heatmaps of mouse (**C**) and human (**D**) Se-associated genes.

**Figure 3 antioxidants-08-00120-f003:**
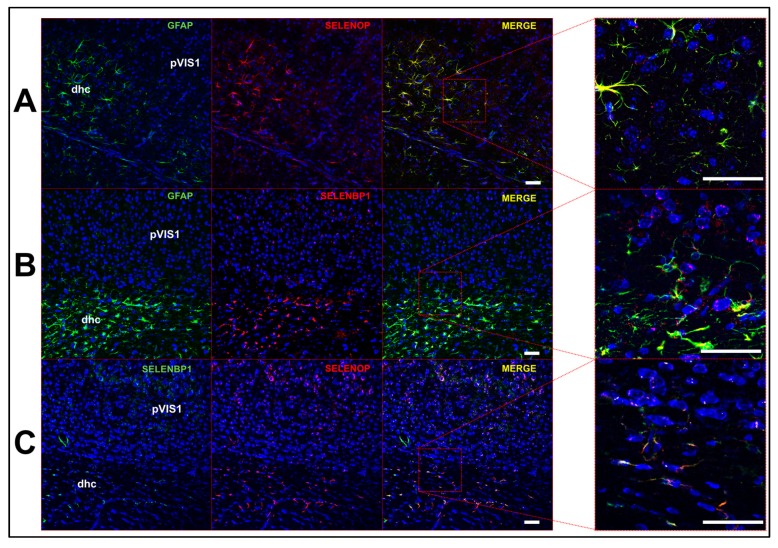
Astrocytes co-express selenoprotein P (SELENOP) and selenium binding protein 1 (SELENBP1). (**A**) Confocal images showing expression of GFAP (left), SELENOP (middle left), and merged images at low (middle right) and high magnification (right). (**B**) Images of GFAP (left), SELENBP1 (middle left), and merged images at low (middle right) and high magnification (right). (**C**) Images of SELENBP1 (left), SELENOP (middle left), and merged images at low (middle right) and high magnification (right). Abbreviations: dhc: dorsal hippocampal commisure; pVIS1: primary visual cortex. Scale bar = 50 μm.

**Figure 4 antioxidants-08-00120-f004:**
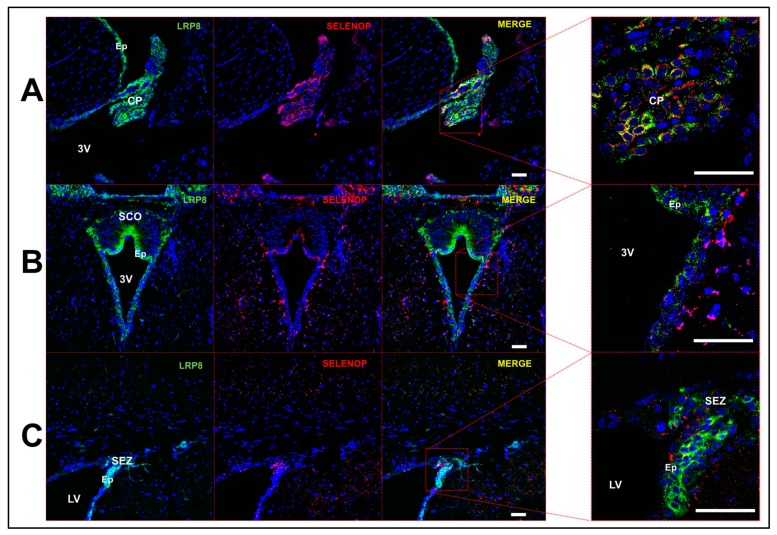
Expression of SELENOP and LRP8 in brain ventricles. (**A**) Confocal images of mouse choroid plexus showing expression of LRP8 (left), SELENOP (middle left), and merged images at low (middle right) and high magnification (right). (**B**,**C**) Images showing expression of LRP8 and SELENOP in the vicinity of the 3rd (**B**) and lateral ventricle (**C**). Abbreviations: 3V: 3rd ventricle; CP: choroid plexus; Ep: ependymal cell layer; LV: lateral ventricle; SEZ: subependymal zone. Scale bar = 50 μm.

**Figure 5 antioxidants-08-00120-f005:**
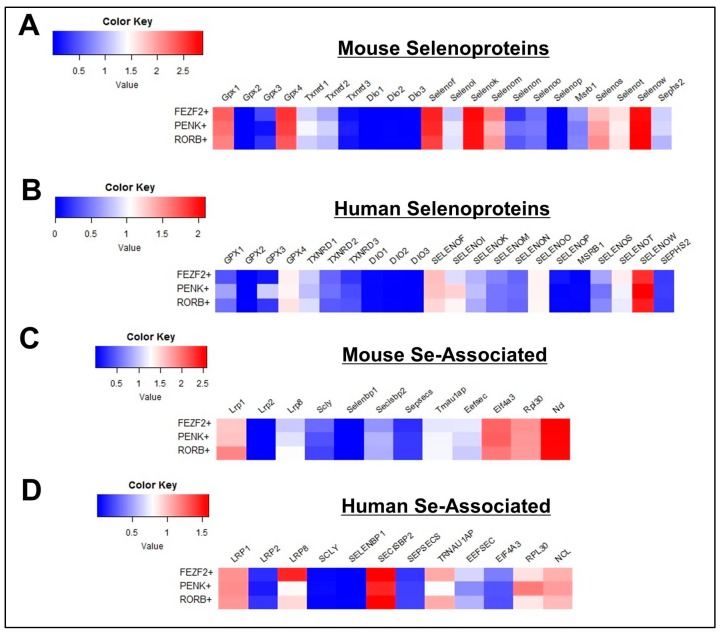
Profile of Se-Related Transcriptome in Glutamatergic Neurons. (**A**,**B**) Heatmaps detailing expression of mouse (**A**) and human (**B**) selenoprotein genes in FEZF2+, PENK+, and RORB+ glutamatergic neurons. (**C**,**D**) Corresponding heatmaps of mouse (**C**) and human (**D**) Se-associated genes.

**Figure 6 antioxidants-08-00120-f006:**
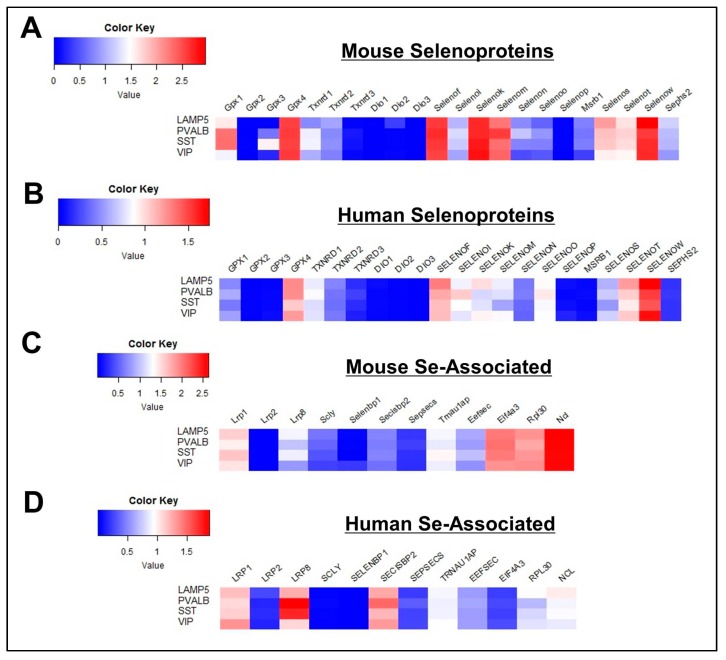
Profile of Se-related transcriptome in GABAergic neurons. (**A**,**B**) Heatmaps detailing expression of mouse (**A**) and human (**B**) selenoprotein genes in LAMP5+, PVALB+, SST+, and VIP+ GABAergic neurons. (**C**,**D**) Corresponding heatmaps of mouse (**C**) and human (**D**) Se-associated genes.

**Figure 7 antioxidants-08-00120-f007:**
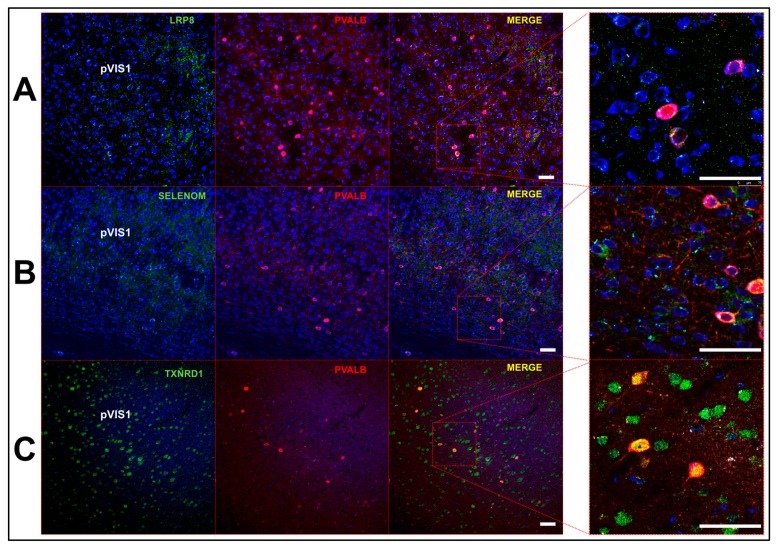
Expression of Se-related proteins in parvalbumin-expressing interneurons (PVIs). (**A**) Confocal images showing expression of LRP8 (left), parvalbumin, (PVALB, middle right) and merged images at low (middle right) and high magnification (right) in the mouse primary visual cortex. (**B**,**C**) Images depicting expression of SELENOM (**B**) and thioredoxin reductases (TXNRD) (**C**) relative to PVALB. Scale bar = 50 μm.

**Figure 8 antioxidants-08-00120-f008:**
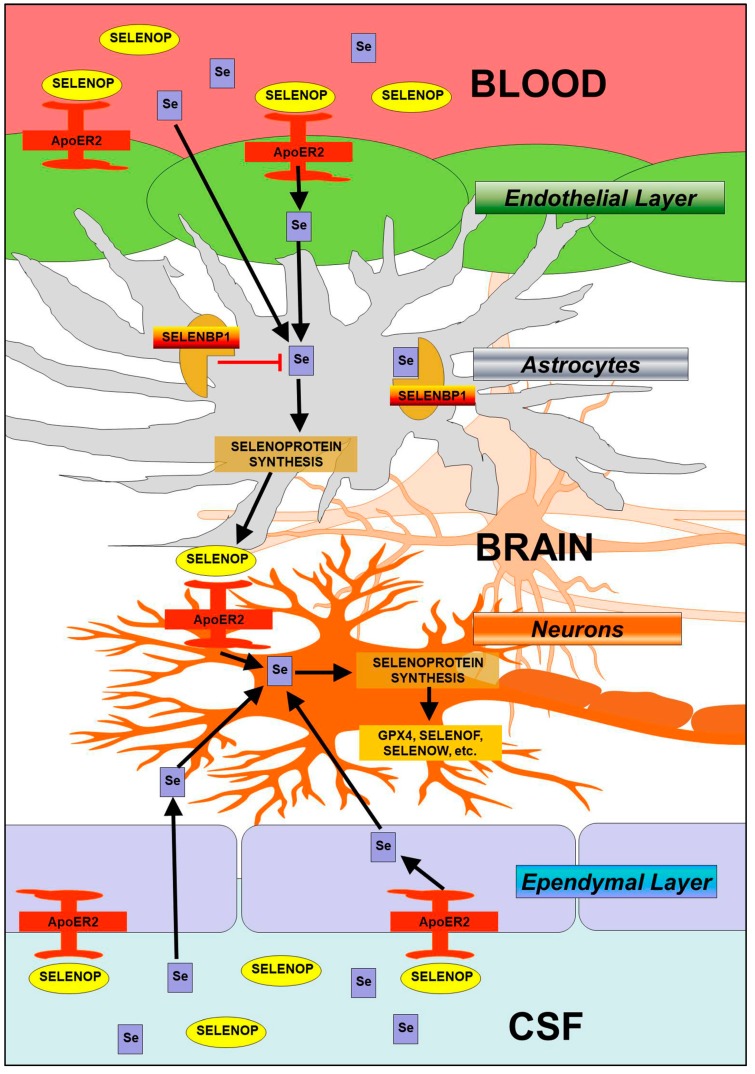
Hypothetical model of Se homeostasis in brain. Circulating SELENOP present in blood and cerebrospinal fluid (CSF) is taken up by LRP8-positive cells in the epithelial (blood–brain barrier) and ependymal (brain–CSF barrier) layers, resynthesized in neighboring astrocytes, and released to supply LRP8-positive neurons with Se. Within astrocytes, SELENBP1 may sequester Se away from selenoprotein synthesis and thereby negatively regulate SELENOP synthesis. Evidence also indicates the existence of parallel, non-SELENOP pathway by which a small molecule form of Se enters the brain.

**Table 1 antioxidants-08-00120-t001:** Cells used to generate data for Figure 1.

Species	Cell Class	Number of Cells
Mouse	Non-Neuronal	1383
Mouse	Glutamatergic	11,905
Mouse	GABAergic	10,534
Human	Non-Neuronal	914
Human	Glutamatergic	10,525
Human	GABAergic	4164

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
