# Peer review of "Cell-Type Specific Analysis of Selenium-Related Genes in Brain"

_antioxidants, 2019, doi:10.3390/antiox8050120_

Round 1

Reviewer 1 Report

The ms entitled Cell-type specific analysis of selenium-related genes in brain” by Alexandru R. Sasuclark et al. provides a wealth of expression data related to genes implicated in selenoprotein metabolism, extracted and edited from a freely available repository.  The results are partly complemented by respective immunohistochemical analyses and nicely highlight potentially relevant differences between the different cell types in brain.

Major: An interpretation of the meaning of the differences observed is missing. What new can we learn from these data, and what novel hypotheses and perspectives can be deduced and experimentally tested in future research? The authors provide one mechanistic interpretation on the role of SELENOP expression for the transport of the trace element in brain, but such ideas should also be put forward for the other findings. An additional figure presenting such potential pathways that are supported by the data would nicely complement this manuscript and attract the attention of the readers.

Minor:

Abstract: “… specific class of antioxidant proteins …“ Selenoproteins are no antioxidants – and some of them have no proven enzymatic activity. Please modify.

Abstract: A conclusion or hypothesis based on the new data presented is needed, to better understand potential implications of these findings.

Abbreviations: GPX, DIO, TXNRD are already plural. Please remove the small “s” at the end.

Introduction: The abbreviation “PVI” needs to become introduced.

Materials and Methods: Information on the number of replicate immunostainings is missing.

Fig. 1: Human genes should be labelled with capital letters, while rodent ones should start with a capital and then followed by small letters.

Fig. 4: very nice.

Discussion: The authors state: “Interestingly, expression levels of SELENOK, SELENOM, and SELENOS were substantially lower in humans relative to that observed in mice, while levels of the poorly characterized SELENOO were elevated.” Why is this finding interesting? Please provide some thoughts on the potential relevance of these data.

“These findings are supportive of a model where SELENOP present in CSF is 274 taken up by LRP8-positive ependymal cells, resynthesized in neighboring astrocytes, and 275 subsequently released to supply LRP8-positive neurons with Se.” This is a fine example of a functional interpretation or speculation on the relevance of the data and the physiological pathway that may underlie these findings. It might be worthwhile to drafting a summarizing scheme including this and other metabolic interpretations from the expression patterns presented as an additional figure.

A recent paper reports on high SELENBP1 serum levels in myocardial infarction and death. Would the authors like to speculate whether brain damage may be related to alterations in SELENBP1 release and re-distribution of Se to or away from the site of injury? The relevance of Se status and selenoprotein expression in brain damage is well established and may be related to reactive changes in the regulation of selenoprotein biosynthesis machinery and Se transport proteins. This notion would fit to the co-expression of the two potential Se-transporting molecules, i.e., SELENOP and SELENBP1 (one mainly extra- and the other intracellularly).

One of the most relevant modulators of selenoprotein expression is the Se status. Can the authors identify any relation of the expression pattern of selenoproteins that may provide an insight into the Se status of the information-providing individual under study?

Another potential relevant difference is sex. From the expression data, the authors should be able to identify male-specific transcripts and thus differentiate between male and female cells. Could the authors please compare for potential differences in expression according to sex, and discuss the findings in light of the vast literature on sex-specific differences in Se biology.

Author Response

1.      Reviewer #1: Major: An interpretation of the meaning of the differences observed is missing. What new can we learn from these data, and what novel hypotheses and perspectives can be deduced and experimentally tested in future research? The authors provide one mechanistic interpretation on the role of SELENOP expression for the transport of the trace element in brain, but such ideas should also be put forward for the other findings. An additional figure presenting such potential pathways that are supported by the data would nicely complement this manuscript and attract the attention of the readers.

Our response: We have added an additional figure detailing a hypothetical model of brain Se homeostasis based upon our results.

2.      Reviewer #1: Abstract: “… specific class of antioxidant proteins …“ Selenoproteins are no antioxidants – and some of them have no proven enzymatic activity. Please modify.

Our response: The language in the abstract has been modified accordingly.

3.      Reviewer #1: Abstract: A conclusion or hypothesis based on the new data presented is needed, to better understand potential implications of these findings.

Our response: An additional sentence has been added to the abstract that puts forth a novel hypothesis for future experimental validation.

4.      Reviewer #1: Abbreviations: GPX, DIO, TXNRD are already plural. Please remove the small “s” at the end.

Our response: The change has been made.

5.      Reviewer #1: Introduction: The abbreviation “PVI” needs to become introduced.

Our response: The abbreviation for PVI is introduced on line 47.

6.      Reviewer #1: Materials and Methods: Information on the number of replicate immunostainings is missing.

Our response: It is now included.

7.      Reviewer #1: Fig. 1: Human genes should be labelled with capital letters, while rodent ones should start with a capital and then followed by small letters.

Our response: Done.

8.      Reviewer #1: Fig. 4: very nice.

Our response: Thank you for your positive feelings.

9.      Reviewer #1: Discussion: The authors state: “Interestingly, expression levels of SELENOK, SELENOM, and SELENOS were substantially lower in humans relative to that observed in mice, while levels of the poorly characterized SELENOO were elevated.” Why is this finding interesting? Please provide some thoughts on the potential relevance of these data.

Our response: We have added a sentence as to why this might be interesting (lines 255-256).

10.     Reviewer #1: “These findings are supportive of a model where SELENOP present in CSF is 274 taken up by LRP8-positive ependymal cells, resynthesized in neighboring astrocytes, and 275 subsequently released to supply LRP8-positive neurons with Se.” This is a fine example of a functional interpretation or speculation on the relevance of the data and the physiological pathway that may underlie these findings. It might be worthwhile to drafting a summarizing scheme including this and other metabolic interpretations from the expression patterns presented as an additional figure.

Our response: Good idea. The schematic is now included as Figure 8.

11.     Reviewer #1: A recent paper reports on high SELENBP1 serum levels in myocardial infarction and death. Would the authors like to speculate whether brain damage may be related to alterations in SELENBP1 release and re-distribution of Se to or away from the site of injury? The relevance of Se status and selenoprotein expression in brain damage is well established and may be related to reactive changes in the regulation of selenoprotein biosynthesis machinery and Se transport proteins. This notion would fit to the co-expression of the two potential Se-transporting molecules, i.e., SELENOP and SELENBP1 (one mainly extra- and the other intracellularly).

Our response: We have revised our language to postulate that intracellular SELENBP1 may serve to sequester Se away from selenoprotein synthesis and thereby limit SELENOP synthesis.

12.     Reviewer #1: One of the most relevant modulators of selenoprotein expression is the Se status. Can the authors identify any relation of the expression pattern of selenoproteins that may provide an insight into the Se status of the information-providing individual under study?

Our response: We do not know the Se status of the individuals used in this study. The mice were likely on standard lab chow (~0.2 ppm Se), but the humans may have varied considerably.

13.     Reviewer #1: Another potential relevant difference is sex. From the expression data, the authors should be able to identify male-specific transcripts and thus differentiate between male and female cells. Could the authors please compare for potential differences in expression according to sex, and discuss the findings in light of the vast literature on sex-specific differences in Se biology.

Our response: We have added a supplemental figure (Figure S1) where male and female data is compiled separately. Also included is a brief summary of these results in the discussion section (lines 315 – 321).

Reviewer 2 Report

Authors reported expression of selenoprotein genes in the mouse and human brain and co-expression of SELENOP and SELENOBP1 in astrocytes. These results could provide a basic evidence for other studies.

Only minor point to be improved is that Table 1-4 are not necessary in the results section. It may present in the method section or it could be combined into one table.  

Author Response

1.    Reviewer #2: Only minor point to be improved is that Table 1-4 are not necessary in the results section. It may present in the method section or it could be combined into one table.  

Our response: We have removed Tables 2 – 4 upon suggestion by Reviewer #2.